# CC Chemokines in a Tumor: A Review of Pro-Cancer and Anti-Cancer Properties of Receptors CCR5, CCR6, CCR7, CCR8, CCR9, and CCR10 Ligands

**DOI:** 10.3390/ijms21207619

**Published:** 2020-10-15

**Authors:** Jan Korbecki, Szymon Grochans, Izabela Gutowska, Katarzyna Barczak, Irena Baranowska-Bosiacka

**Affiliations:** 1Department of Biochemistry and Medical Chemistry, Pomeranian Medical University in Szczecin, Powstańców Wielkopolskich 72 Av., 70-111 Szczecin, Poland; jan.korbecki@onet.eu (J.K.); szymongrochans@gmail.com (S.G.); 2Department of Medical Chemistry, Pomeranian Medical University in Szczecin, Powstańców Wlkp. 72 Av., 70-111 Szczecin, Poland; izagut@poczta.onet.pl; 3Department of Conservative Dentistry and Endodontics, Pomeranian Medical University, Powstańców Wlkp. 72 Av., 70-111 Szczecin, Poland; kasiabarczak@vp.pl

**Keywords:** chemokine, CC chemokine, cancer, tumor, organ-specific metastasis, angiogenesis, lymphangiogenesis, tumor microenvironment

## Abstract

CC chemokines (or β-chemokines) are 28 chemotactic cytokines with an N-terminal CC domain that play an important role in immune system cells, such as CD4^+^ and CD8^+^ lymphocytes, dendritic cells, eosinophils, macrophages, monocytes, and NK cells, as well in neoplasia. In this review, we discuss human CC motif chemokine ligands: CCL1, CCL3, CCL4, CCL5, CCL18, CCL19, CCL20, CCL21, CCL25, CCL27, and CCL28 (CC motif chemokine receptor CCR5, CCR6, CCR7, CCR8, CCR9, and CCR10 ligands). We present their functioning in human physiology and in neoplasia, including their role in the proliferation, apoptosis resistance, drug resistance, migration, and invasion of cancer cells. We discuss the significance of chemokine receptors in organ-specific metastasis, as well as the influence of each chemokine on the recruitment of various cells to the tumor niche, such as cancer-associated fibroblasts (CAF), Kupffer cells, myeloid-derived suppressor cells (MDSC), osteoclasts, tumor-associated macrophages (TAM), tumor-infiltrating lymphocytes (TIL), and regulatory T cells (T_reg_). Finally, we show how the effect of the chemokines on vascular endothelial cells and lymphatic endothelial cells leads to angiogenesis and lymphangiogenesis.

## 1. Introduction: The Dual Properties of Chemokines Are Key for Understanding of the Tumor Microenvironment during Therapy

The CC (β) subfamily of chemokines is a group of chemotactic cytokines known as CC motif chemokine ligands (CCL)1–28. Their shared characteristic is the N-terminal CC domain and digits in their symbols depend on the order of discovery [1,2]. The actual number of CC chemokines is 27, as CCL9 and CCL10 denote the same chemokine. All these chemokines are ligands for 10 receptors—CC motif chemokine receptors (CCR)1–10. Just like the rest of chemokines, CC chemokines are crucial for the functioning of the immune system cells [2]. However, apart from their anti-cancer properties, they also show some pro-cancer characteristics and thus play an important role in neoplasia.

Below, we have listed the properties of selected chemokines in the tumor. These properties can be divided into anti- and pro-cancer. The former are associated mainly with the recruitment of anti-cancer tumor-infiltrating lymphocytes (TIL), which destroy cancer cells [3]. In turn, the pro-cancer properties of chemokines are related to recruiting cells supporting tumor development [4], as well as increasing or causing the proliferation [5], migration, and invasion of cancer cells [6]. Very often, if not always, a given chemokine shows both pro- and anti-cancer properties, something we wished to emphasize in this paper. Therefore, the increased expression of a single chemokine is not always a clear indicator in establishing a patient prognosis in all cancers. The table below shows where the increased expression of a given chemokine improves the prognosis for one type of cancer or worsens the prognosis for patients with other cancers [7,8] (Table 1 and Table 2). There is no single CC chemokine that would worsen or improve the prognosis in all types of cancers.

Another important premise of this review is the intratumor heterogeneity. A tumor is not a homogenous environment and consists of areas with different properties. The most significant is the area affected by chronic hypoxia [9], characterized by the accumulation of tumor-associated macrophages (TAM) [10,11,12,13], regulatory T cells (T_reg_) [14,15,16], and myeloid-derived suppressor cells (MDSC) [17,18]. The functions of these recruited pro-cancer cells in this microenvironment are enhanced by chronic hypoxia [11,19,20,21] and cancer acidification [22], which increases the resistance of cancer cells to anticancer therapy and the action of the immune system [20,22,23,24]. In such hypoxic areas, chemokines show only pro-cancer properties, despite their aforementioned dual nature. However, during the effective anti-cancer response of the immune system, the same chemokines will exhibit anti-cancer properties [3] (Figure 1).

Knowledge of the anti-cancer and pro-cancer properties of individual chemokines allows a prediction of the consequences to then improve the effectiveness of anti-cancer therapies. One example is radiotherapy, which leads to an increased expression of certain chemokines, e.g., CCL2 and CCL5, resulting in the recruitment of TAM and T_reg_ [25,26,27]. This has a pro-cancer effect and nullifies the therapeutic benefits of radiotherapy. On the other hand, the same chemokines have anti-cancer properties, because they infiltrate the tumor with anti-cancer TILs [3,28]. For this reason, radiotherapy may be more effective if used prior to immunotherapy [29]. Another application of knowledge presented in this paper is the use of gene therapy to enhance the expression of a given chemokine followed by immunotherapy [30,31,32,33] or chemotherapy [34]. As in the previous case, the increased expression of a chemokine may enhance the effectiveness of immunotherapy.

The role of chemokines in cancer has been the subject matter of a considerable number of papers. PubMed (https://pubmed.ncbi.nlm.nih.gov/) includes almost 22 thousand articles containing the words “chemokine” and “cancer” (title + abstract + keywords). However, the sheer number of papers makes it difficult to write a comprehensive and meaningful review. One solution is to focus on literature reviews that deal with individual chemokines, although one problem is that due to the fast advances in science they quickly become out-of-date. Another problem is the low attention paid in literature to less important chemokines and the emphasis on the best known chemokines. That is why, in this review, we address the significance of all human chemokines in cancer. Due to the considerable amount of data, we decided to discuss in this paper only CC chemokines that have been assigned to receptors: CCR5, CCR6, CCR7, CCR8, CCR9, and CCR10 (CCL1, CCL3, CCL4, CCL5, CCL18, CCL19, CCL20, CCL21, CCL25, CCL27, and CCL28) (Table 3 and Table 4).

## 2. CCR5 Ligands

### 2.1. CCL5

Chemokine CCL5 (also known as regulated on activation, normally T cell expressed and secreted (RANTES)) is a ligand for receptors CCR5 [35,36], CCR3 [37,38,39], and CCR1 [1,2,40]. After secretion of this chemokine outside the cell, two amino acids are truncated from the N-terminus by CD26/dipeptidyl peptidase IV (DPPIV), resulting in an increased affinity to CCR5 and reduced affinity to CCR1 and CCR3 [41]. CCL5 is also a ligand for G-protein-coupled receptor 75 (GPR75), whose expression occurs on brain nerve cells [42,43]. The effect of CCL5 is mitigated by the atypical chemokine receptor 2 (ACKR2)/D6, which reduces the level of this chemokine [44,45,46]. 

Pro-inflammatory cytokines elevate CCL5 expression, which triggers the accumulation of various immune cells at inflammatory sites [47], e.g., contributing to asthma by CCR3–mediated chemotaxis of eosinophils [47]. The CCL5→CCR5 axis also induces the accumulation of and increases the cytotoxic properties of anti-cancer TIL in a tumor [28,30,48,49,50,51,52,53,54]. CCL5 is responsible for the infiltration of a tumor by NK cells [48,49,52,53], conventional type 1 dendritic cells [54], T helper cell type 1 (Th1) [28], and type 1 cytotoxic cells (Tc1) [22]. For this reason, some postulate that cancer therapies should combine immunotherapy with increasing the expression of CCL5 in the tumor to enhance the infiltration of the tumor by various immune cells [30,49,50]. On the other hand, the genome of Kaposi’s sarcoma-associated herpesvirus 8 contains viral macrophage-inflammatory protein-II (vMIP-II) [55,56,57]. Cells infected by this virus secrete vMIP-II, which acts as a CCR5 antagonist and therefore reduces the infiltration of the tumor by anti-cancer TIL. vMIP-II also induces the recruitment of Th2-type cells via CCR3 and CCR8, which decreases the activity of cytotoxic lymphocytes [58,59]. 

On the other hand, CCL5 has also several pro-cancer properties (Figure 2). Its expression is elevated in breast cancer [60], glioblastoma multiforme [61], and hepatocellular carcinoma [62]. In a tumor, CCL5 expression is found in cancer cells [63,64], cancer-associated fibroblasts (CAF) [65], mesenchymal stem cells (MSC) [66], MDSC [67], TAM [68,69,70,71], and TIL [72]. CCL5 expression also occurs in lymphatic endothelial cells, which is crucial for the formation of the metastatic niche [73]. 

CCL5 acts on cancer cells by increasing their proliferation [62,64,74,75]. By activating the Wnt/β-catenin→signal transducer and activator of transcription 3 (STAT3) signaling pathways, it induces the self-renewal of prostate cancer stem cells [70]. Similar to other chemokines, CCL5 causes migration, invasion, and epithelial-to-mesenchymal transition (EMT) of cancer cells via its receptor [62,64,76,77], which is related to the activation of hedgehog [78], Wnt/β-catenin [70,79], and the Akt/PKB→nuclear factor κB (NF-κB) pathways [80,81,82]. The induction of migration by CCL5 is associated with increased secretion of matrix metalloproteinases (MMPs) by cancer cells [80,83,84]. In addition, CCL5 increases apoptosis resistance and drug resistance through the activation of Akt/PKB→NF-κB and STAT3 pathways [72,85,86,87]. It also causes an increase in programmed death-ligand 1 (PD-L1) expression on cancer cells, which protects them against cytotoxic lymphocytes [71]. 

CCL5 not only affects cancer cells but also tumor-associated cells. It recruits MDSC [88,89], MSC [90], TAM [77,91,92], and T-helper cells 17 (Th17) [93] into the tumor niche. Via CCR5, it participates in recruiting T_reg_ and elevating its activity in the tumor niche [67,94,95,96]. CCL5 is secreted by diffuse large B cell lymphoma, which causes the recruitment of monocytes that increase the proliferation of those cancer cells [97]. In human papillomavirus (HPV) 16, E7 protein increases the expression of CCL5 [98]. This chemokine is responsible for recruiting mast cells and through these cell enhances the growth of a tumor.

CCL5 participates in angiogenesis, which is associated with an increase in vascular endothelial growth factor (VEGF) expression in cancer cells and vascular endothelial cells via the activation of CCR1 and CCR5 [99,100,101,102]. CCL5 may cause differentiation of cancer stem cells into endothelial cells as observed in the ovarian cancer model [103]. This chemokine is also induced by the Epstein–Barr virus (EBV) (human herpesvirus 4) in nasopharyngeal carcinoma cells, which leads to angiogenesis [104]. It may also indirectly cause lymphangiogenesis by increasing VEGF-C expression, as observed in chondrosarcoma cells [105].

### 2.2. CCL3 and CCL4

CCL3 (also known as macrophage inflammatory protein-1α, i.e., MIP-1α) and CCL4 (also known as macrophage inflammatory protein 1β, i.e., MIP-1β) are pro-inflammatory chemokines. In humans, there are also additional copy number variations (CNV) of *CCL3* and *CCL4* genes, known as *CCL3L* and *CCL4L*, which are produced by duplicating the genes of the corresponding chemokines [106,107]. The products of these additional genes have the same amino acid sequence as CCL3 and CCL4 and the same properties. 

CCL3 is a ligand for receptors CCR1 and CCR5 [35,36], while CCL4 interacts with CCR5 [35,36], and with low affinity with CCR1 [40]. CCL3 and CCL4 are not ligands for CCR3 [38,39,108], but Combadiere et al. show that CCL3 and CCL4 are potent agonists for CCR3 [37]. Due to their anti-cancer properties, we decided to include them in the section on CCR5 ligands. Both these CC chemokines are important in the onset of the immune response [109]. They induce the recruitment of dendritic cells, neutrophils, monocytes, macrophages, NK cells, and T cells to inflammatory sites [109]. CCL3 is responsible for the correct function of CD8^+^ T cells and CCL4 acts on CD4^+^ T cells [110,111,112]. The increase in the number of immune cells, induced by CCL3 and CCL4, makes these chemokines a potentially important element of cancer immunotherapy [113,114,115]. 

In a tumor, an active immune response results in the production of CCL3 and CCL4 by B cells [116] and basophils [117]. The produced chemokines act as chemoattractants for anti-cancer TIL with CCR1 and CCR5 [115,118,119,120,121]. However, in a tumor both CCL3 and CCL4 may be cleaved by cathepsin D [122] and chemokine decoy receptor ACKR2/D6 [2,123], which suppresses the anti-cancer effect of these two CC chemokines. 

On the other hand, CCL3 and CCL4 also have pro-cancer effects. Their expression has been found in tumor-associated and cancer cells. In a tumor, they are produced by TAM [69,124,125,126,127], MDSC [68], and MSC [66], which promotes tumor growth. The expression of CCL4 may also occur in tumor-associated neutrophils (TAN) [128]. CCL3 may be expressed in a cancer cell [127]. The immune response reduces the expression of CCL3 in cancer cells via the activation of β-catenin [129]. Both CCL3 and CCL4 are also expressed in chronic myeloid leukemia and multiple myeloma [130,131,132,133]. CCL3 and CCL4 lead to the inhibition of normal osteoblast function and an increase in osteoclast activity and thus to bone destruction [130,134,135,136]. In addition, the production of CCL3 by acute myeloid leukemia cells causes severe anemia by inhibiting erythropoiesis [137]. In bone marrow, hypoxia induces an increase in CCR1 expression on multiple myeloma cells [138]. The activation of this receptor by CCL3 affects the function of CXC motif chemokine receptor 4 (CXCR4), which lead to the egress of those cells to the blood. The elevated level of CCL3 in serum is associated with a worse prognosis for patients with diffuse large B cell lymphoma [139], extranodal NK/T-cell lymphoma [140], and multiple myeloma [141].

Both chemokines can cause T_reg_ recruitment via CCR1 and CCR5. CCL3 causes the recruitment of T_reg_ to the leukemic hematopoietic microenvironment, which promotes the development of acute myeloid leukemia [142]. CCL4 also induces the recruitment of T_reg_ to the tumor niche, as shown in melanoma [67]. Both CCL3 and CCL4 cause the recruitment of MDSC into the tumor niche via CCR5 [143,144]. 

There are no reports on the recruitment of TAM by CCL3 or CCL4; they support the anticancer functions of monocytes [145], so they will not recruit TAM but will be more responsible for the infiltration of anti-cancer M1 macrophages. In a hepatocellular carcinoma model, CCL3—via CCR1—causes the recruitment of Kupffer cells, resident macrophages in the liver [146]. In addition, CCL3 [147] and CCL4 [148] cause the recruitment of CAF via CCR5, which is of great importance for the functioning of a tumor and the initial stages of metastatic niche formation in bones. 

Although CCL3 and CCL4 have an anti-cancer effect by causing cytotoxic TIL infiltration to the tumor, they can support the development of the tumor if they act directly on a cancer cell. For example, CCL3 increases cancer cell proliferation [144]. In multiple myeloma, this chemokine causes drug resistance by activating the PI3K→Akt/PKB→mTOR and extracellular signal-regulated kinase (ERK) mitogen-activated protein kinase (MAPK) pathways [133]. In acute lymphoblastic leukemia, anticancer drugs increase the expression of CCL3 and CCL4, which leads to drug resistance of these cells [149].

CCL3 causes migration and invasion of cancer cells via CCR5 [124,127,144]. This is related, among other things, to the activation of PI3K→Akt/PKB and ERK MAPK pathways [127]. Both CCL3 and CCL4 may also participate in metastasis. CCL3 is secreted by the tumor, which causes an increase in CCL2, CCL7, and CCL8 expression in the lungs and brain [150], which facilitates the metastasis to these organs. After getting to the blood, a cancer cell is encased in platelets, which protects it against NK cells [151]. Platelets also secrete many factors, including CCL3, which support the cancer cell [152]. After reaching another organ, CCL3 participates in the formation of the metastatic niche. The cancer cell recruits macrophages through the CCL2→CCR2 axis [153]; these macrophages then secrete CCL3, which causes the retention of these cells in the metastatic niche in an autocrine manner via CCR1. In bones, on the other hand, there is an increase in the expression of CCL3 in bone-marrow-derived monocytes caused by the secretion of epidermal growth factor (EGF) by cancer cells. This causes the differentiation of these monocytes into osteoclasts [154], which leads to bone remodeling around the forming metastases. 

CCL3 and CCL4 also indirectly cause angiogenesis. CCL3 [127,155] and CCL4 [156] cause increased expression of VEGF in a cancer cell, a growth factor causing angiogenesis. CCL4 also increases the expression of VEGF-C in a cancer cell, which leads to lymphangiogenesis and lymph node metastasis [157]. 

## 3. CCR6 and CCL20

CCL20 (also known as liver activation regulated chemokine (LARC) or macrophage inflammatory protein-3α (MIP-3α) or Exodus-1) is a proinflammatory chemokine; a ligand for CCR6 [1,2,158,159]. This makes it significant for the correct function of dendritic cells, T cells, and B cells—cells with expression of this receptor [159]. CCL20 is produced by Th17 cells and is thus responsible for the function of these lymphocytes [160]. In addition, CCL20 also plays an important role in neoplastic processes (Figure 3). 

CCL20 expression is elevated in many cancers, such as breast cancer [60], hepatocellular carcinoma [161], and pancreatic cancer [162,163]. While its expression occurs in cancer cells [63,162], TAM are also a significant source of this chemokine in the tumor [162,164,165]. CCR6 expression has also been shown on tumor cells. Thus, in some cancers, the cells can stimulate their own proliferation and migration in an autocrine manner via CCL20→CCR6 [63,166,167,168]. At the same time, CCL20 directly causes angiogenesis by activating CCR6 on vascular endothelial cells [169]. CCR6 activation may also increase the expression of VEGF in cancer cells, which contributes to angiogenesis [170]. 

The main primary function of CCL20 in a tumor is to recruit T_reg_ and Th17 for the tumor niche [171,172,173,174]. This leads to tumor immune evasion. The CCL20 also recruits TAMs [175] through CCR6. This chemokine is also responsible for recruiting dendritic cells, which increases the anticancer response of the immune system [176]. However, it seems that the impact of cancer immune evasion is stronger than this process. 

In addition to recruiting cells to a tumor niche, CCL20 increases cancer cell proliferation [166,177], causes cancer cell migration and invasion [167,177,178,179], and induces EMT of cancer cells [165,180,181]. It also participates in organ-specific patterns of metastasis—high expression of CCL20 in the liver results in the metastasis of tumor cells with high expression of CCR6 to this organ [182,183,184]. CCR6 is also important in lung metastasis in breast cancer patients [185] and adrenal metastasis in lung cancer patients [186]. However, in the adrenal gland and lungs, CCL20 expression is low [187]. This indicates that this chemokine can only participate in the induction of cell migration into the bloodstream. The expression of CCR6 on B-cell non-Hodgkin’s lymphomas leads to the localization of those cells at mucosal sites [188]. The CCL20→CCR6 axis is also significant in osteolytic bone lesions caused by multiple myeloma [189].

## 4. CCR7, CCL19, and CCL21

The most important physiological functions of CCL19 (also known as EBI1 ligand chemokine (ELC) or macrophage inflammatory protein-3 (MIP-3) or Exodus-3) and CCL21 (also known as secondary lymphoid tissue chemokine (SLC) or 6Ckine or Exodus-2) include the homing of T cells to a lymph node [190,191,192,193,194]. This process is dependent on the receptor of these chemokines: CCR7 [2,190]. Therefore, the increased expression of these chemokines in a tumor has an anti-cancer effect via cytotoxic TIL [195,196,197,198]. 

However, this axis may also have pro-cancer properties if CCR7 is located on a cancer cell. Hypoxia [199,200,201] and prostaglandin E_2_ (PGE_2_) [202,203] increases the expression of CCR7 on a cancer cell (Figure 4). CCR7 enhances proliferation [204,205,206] and stemness of cancer cells [207,208,209]. The activation of CCR7 increases angiogenesis by activating NF-κB and thus increases VEGF-A expression in esophageal squamous carcinoma cells [210]. In contrast, in colorectal cancer cells, CCL19 inhibits angiogenesis by increasing the expression of miR-206, which inhibits the ERK MAPK→HIF-1→VEGF-A pathway [211]. Activation of CCR7 also increases the expression of VEGF-C and VEGF-D in non-small-cell lung cancer cells [210], head and neck cancer cells [212], and esophageal squamous carcinoma cells [213]. These are growth factors responsible for lymphangiogenesis. The activation of CCR7 on a cancer cell causes EMT and migration of cancer cells [199,200,204,205,206,209]. In blood and lymphatic vessels CCR7 on cancer cells prevents anoikis [205,214]. In lymphangiogenesis, cancer cells penetrate lymphatic vessels where a cancer cell with CCR7 expression is kept in a lymph node [215,216,217,218,219,220,221], which is related to the high expression of a ligands for this receptor [190,191,192,194] in these peripheral lymphoid organs. In addition, some leukemias, such as B cell chronic lymphocytic leukemia but not multiple myeloma cells, show CCR7 expression [222]. This leads to the homing of those cells to secondary lymphoid tissues. CCR7 is also significant in the dissemination of non-Hodgkin’s lymphoma [223]. In leukemias, the expression of CCR7 is a significant prognostic factor. Higher expression is associated with worse prognosis for patients with diffuse large B-cell lymphoma [224]. Metastasis to lymph nodes is not only associated with the described mechanism but also with the high expression of CCL1 in a lymph node combined with the expression of CCR8 on a cancer cell [225]. This process is crucial for the metastasis of malignant melanoma to a lymph node. CCR7 expression has also been associated with skin metastasis in patients with breast cancer [185].

In addition to the effect on a cancer cell, the described axis CCL19/CCL21→CCR7 also affects the cellular composition of a tumor. Increased expression of CCL19 and CCL21 in a tumor results in anti-cancer TIL infiltration and an improved prognosis for many tumor patients [195,196,197,198]. Cancer immune evasion mechanisms reduce the expression of these chemokines in a cancer cell [129]. Nevertheless, if the tumor microenvironment has favorable conditions for tumor development, then CCL19 and CCL21 may participate in T_reg_ recruitment to the tumor niche [226,227]. T_reg_ are cells involved in cancer immune evasion and so their recruitment to a tumor niche inhibits the correct anti-cancer response of the immune system.

## 5. CCR8

### 5.1. CCL1

Chemokine CCL1(also known as I-309) was first identified as a cytokine secreted by activated human T lymphocytes [228]. CCL1 is a ligand for just one receptor, CCR8 [1,2,229], and is considered a Th2-related cytokine due to the expression of the CCR8 receptor on Th2 cells, but not on Th1 cells [230]. Therefore, this chemokine plays an important role in the pathogenesis of asthma [231]. The CCL1→CCR8 axis also plays an important role in the homing of lymphocytes to the healthy skin and therefore plays an important role in the physiology of this tissue [232,233].

Elevated expression of CCL1 in a cancer cell occurs in leukemia caused by viruses, e.g., human T cell leukemia virus type 1 (HTLV-1) [234,235]. Properties similar to this chemokine are also shown by viral macrophage inflammatory protein-I (vMIP-I) (homolog to CCL1)—a viral protein expressed in tumor cells transformed by the herpes virus [236]. Another example is Kaposi sarcoma-related human herpes virus-8 [237,238]. In addition, encoded in by the genome of that virus, vMIP-II leads to the recruitment of Th2-type cells via CCR3 and CCR8, which reduces the activity of cytotoxic lymphocytes [58]. In breast cancer, however, the expression of CCL1 is no different than in healthy tissue [60]. In hepatoma cells, it is absent or very low [239]. In gliomas, on the other hand, it is even lower than in the brain tissue [61]. In hepatocellular carcinomas, the expression of CCL1 is elevated in tumor stroma and peritumoral tissue [240]. In solid tumors, CCL1 is produced by CAF [241,242], TAM [243], CCR8^-^CD11b^+^ myeloid cells [244], and T_reg_ [245]. The expression of this chemokine has also been demonstrated in breast cancer stem cells [246], bladder cancer tumors [244,247], and renal cell carcinomas [244]. 

In cancer, CCL1 has an antiapoptotic activity and induces chemoresistance to anticancer drugs due to the activation of the ERK MAPK cascade via CCR8 (Figure 5) [234,236,242,248,249]. This has a special significance in apoptosis resistance adult T-cell leukemia [234] and murine T cell lymphomas [248]. CCL1 also stimulates proliferation [247] and causes the migration of bladder cancer cells [241]. At the same time, due to the expression of CCL1 in lymph nodes, this chemokine participates in metastasis to these peripheral lymphoid organ through the CCR8 receptor on cancer cells that have entered lymphatic vessels [225]. This process is important in metastasis to lymph nodes in malignant melanoma, which often leads to increased CCR8 expression in the cancer cells of this tumor [225]. However, the CCL1→CCR8 axis is not the only molecular route of metastasis to lymph nodes. In this process, CCL19 and CCL21 acting on CCR7 are also significant [215,216,217,218,219,220,221]. 

In addition to the effect on a cancer cell, CCL1 causes angiogenesis on vascular endothelial cells via CCR8 [238,250]. It is involved in the recruitment of CCR8^-^CD11b^+^ myeloid cells [244] and T_reg_ [246,251,252] to a tumor niche. However, CCL1 can also participate in the conversion of CD4^+^ T cells into T_reg_ [245], in a process dependent on transforming growth factor-β (TGF-β), which increases the expression of CCL1 in CD4^+^ T cells. This is followed by an autocrine conversion of this cell to T_reg_, which also involves CCL1. In addition, CCL1 supports the immunosuppressive function of T_reg_ in a tumor niche [245], which is crucial for the interaction of cancer stem cells and CAF with T_reg_ [241,242,246]. Finally, CCL1 increases the expression of interleukin 6 (IL-6) in MDSC, which acts as a proinflammatory in the cancer microenvironment [244].

### 5.2. CCL18

CCL18 (also known as pulmonary and activation regulated chemokine (PARC), alternative macrophage activation-associated C-C chemokine-1 (AMAC-1), dendritic cell-derived C-C chemokine 1 (DC-CK1), and macrophage inflammatory protein 4 (MIP-4)) is produced by dendritic cells, especially in germinal centers of regional lymph nodes. This chemokine then causes the chemoattraction of naïve T cells to these cells [253,254,255]. This leads to the initiation of an immunological response. CCL18 also acts as an anti-inflammatory chemokine, a marker for the macrophage M2 subset [256]. 

In a tumor, CCL18 is mainly produced by TAM (Figure 6) [69,257,258]. This chemokine causes cancer cell migration and EMT by activating its receptors, phosphatidylinositol transfer protein 3 (PITPNM3) and CCR8 [2,259,260,261,262,263]. CCL18 also increases the expression of cancer stem cell markers [264], is involved in angiogenesis by acting on vascular endothelial cells via PITPNM3 [265], and affects non-cancer cells in the tumor niche. Acting as an immunosuppressive cytokine, it causes polarization of macrophages to phenotype M2 [256] and recruits naïve CD4^+^ T cells into the tumor niche via its receptor PITPNM3 [266], which then differentiate into T_reg_ cells responsible for tumor immune evasion. CCL18 recruits immature dendritic cells to the tumor niche [267]. In addition, this chemokine participates in the differentiation of immature dendritic cells into tumor-associated dendritic cells (TADC) [268,269]. Other functions of CCL18 in the tumor include participation in the intercellular communication dependent on extracellular vesicles [270]. This chemokine binds to glycosaminoglycans on extracellular vesicles, which allows them to be retained on cells with an expression of CCR8, a receptor for CCL18. CCL18 also increases the proliferation of cancer cells, but this effect depends on the type of tumor. For example, CCL18 decreases the proliferation of acute lymphocytic leukemia B cells [271] and cutaneous T-cell lymphoma (CTCL) [272] in a process dependent on GPR30 that affects the activity of CXCR4 [271]. Increased expression of CCL18 in lesional skin and serum of patients with CTCL [273] and in patients with diffuse large B cell lymphoma [274] is associated with a worse prognosis. In addition, CCL18 decreases the proliferation of non-small-cell lung cancer cells [67] and increases the proliferation of glioma cells [275] and oral squamous cell carcinoma cells [276].

## 6. CCR9 and CCL25

CCL25 (also known as thymus-expressed chemokine (TECK)) is a ligand for just one receptor: CCR9 [1,2,277,278]. This chemokine is important for the correct function of the thymus [279,280]. Due to the expression of CCL25 in the gastrointestinal tract, lymphocyte homing to the tissues of this system is one of the main functions of the CCL25→CCR9 axis [281,282,283]. This is of great importance in the immunological functions of the intestinal and gastric mucosa. However, this function of CCL25 results in intestinal metastasis of cancer cells with CCR9 expression [284,285]. The CCL25→CCR9 axis is also significant in the homing of diffuse large B-cell lymphoma and follicular lymphoma to the gastrointestinal tract [286]. 

In a tumor, CCL25 is produced by cancer cells, such as breast cancer cells [287] and pancreatic cancer cells [288], and also by cancer-related cells, e.g., pancreatic stellate cells in pancreatic cancer [289]. CCL25 participates in the migration and invasion of cancer cells by causing an increase in metalloproteinase expression [290,291,292]. It also causes EMT of cancer cells [287]. However, in colon cancer, CCR9 activation inhibits cancer cell migration [293]. The activation of CCR9 is also associated with apoptosis resistance and drug resistance, which is related to Akt/PKB→PI3K pathway activation [294,295,296]. Additionally, the CCL25→CCR9 axis stimulates tumor cell proliferation [288]. 

There are no data on the influence of CCL25 on the recruitment of cells cooperating in the development of a tumor or its participation in angiogenesis [297]. However, this chemokine may cause infiltration of the tumor by cytotoxic TIL exhibiting CCR9 expression, which has an anticancer effect [298]. There are also indications that CCL25 causes lymphangiogenesis because activation of the CCL25→CCR9 axis increases the expression of VEGF-C and VEGF-D on the non-small-cell lung cancer cells [292]. CCL25 can also recruit MDSC into a tumor niche, as indicated in a study by Sun et al. on endometriosis [299].

## 7. CCR10, CCL28, and CCL27

CCL28 (also known as mucosae-associated epithelial chemokine (MEC)) is a chemokine essential for normal mucosal immune function [300,301]. It activates two receptors: CCR3 [302] and CCR10 [303]. Another chemokine that activates CCR10 is CCL27 (also known as ESkine) [2,304,305], and for this reason, its effects are similar to those of CCL28. CCL27 causes CD3^+^ and CD4^+^ lymphocyte migration [306]. CCL28 and CCL27 cause B cell and T cell migration via CCR10, especially IgA plasma blasts [300,301,307]. CCL27 is produced by keratinocytes and therefore acts mainly in the skin [301], whereas CCL28 is produced in mucosal tissues [301]. For this reason, these chemokines are important in the homing of immune system cells to mucosal and epithelial tissues and thus participate in immunological reactions against microorganisms. CCL28 also causes eosinophil migration via CCR3 [302], which is associated with the development of allergies. 

CCL27 also has an intracellular function. The CCL27 gene, through alternative splicing, creates—in addition to CCL27—PESKY protein [306,308,309], a nuclear protein with expression in the eye, brain, and testes. This protein alters the expression of genes associated with the actin cytoskeleton, which leads to cell migration and changes in cell morphology [308,310]. CCL27 also contains a nuclear location sequence—after internalization of the CCR10 receptor with this chemokine, CCL27 is transported to the cell nucleus where it has a similar function to PESKY [308]. 

Expression of CCL28 is down-regulated in breast cancer [311], colon cancer [312], and multiforme glioblastoma [61]. In endometrial cancer [313], basal cell carcinoma [314], and squamous cell carcinoma [314], the expression of CCL27 is reduced. This suggests that these chemokines have an anticancer effect, at least in the early stages of tumor development. This is confirmed by studies on the survival of patients with different types of breast cancer [315]. In patients with luminal-like breast cancer, elevated concentrations of CCL28 in the tumor improve prognosis, in contrast to triple-negative breast cancer [315]. In turn, elevated CCL27 expression improves the prognosis in patients with cutaneous malignant melanoma [316]. CCL28 and CCL27 participate in the anticancer response of the immune system, causing an infiltration of the tumor by anticancer NK cells, which leads to improved prognosis with an increased expression of these chemokines in the tumor [313,317,318,319]. For this reason, gene therapies that increase the expression of these two chemokines have an anticancer effect [317,318,320]. 

However, numerous in vitro studies show that CCL28 and CCL27 support tumor development (Figure 7). For this reason, some researchers suggest that CCL28 can act locally, especially in the hypoxic regions of the tumor [14,16,17,321,322]. CCL28 and CCL27 stimulate proliferation and have anti-apoptotic effects on cancer cells [323,324,325]. In vitro experiments also show that CCL28 reduces migration and EMT in oral squamous cell carcinoma [326], but the effect on migration varies among other types of cancer. In hepatocellular carcinoma, CCL28 increases the migration of these cells [327]. The same activation of CCR10 by both the ligands causes the migration of breast cancer cells [325,328] and glioblastoma multiforme cells [324]. After entering the bloodstream, a cancer cell is trapped in organs and tissues, which have a high expression of chemokines for the receptors on the cancer cell. Due to the production of CCL28 and CCL27 in the skin, the expression of CCR10 on a cancer cell increases the probability of skin metastasis [329].

In addition to its effect on cancer cells, CCL28 also acts on non-cancer cells in a tumor niche. In particular, it participates in the recruitment of T_reg_ [14,16], and in pancreatic ductal adenocarcinoma, in the recruitment of cancer-associated stellate cells [330] caused by the expression of CCR10 on these cells. The CCL27→CCR10 axis has also been shown to be involved in the recruitment of Th22 to malignant ascites; even though no effect of an increased number of these cells on patient prognosis has been demonstrated [331], Th22 may have a pro-cancer effect, as shown in a study on colorectal cancer [332]. In turn, CCL28 causes angiogenesis by activating CCR3 on vascular endothelial cells [322]. Both chemokines, CCL28 and CCL27, cause lymphangiogenesis by activating CCR10 on lymphatic endothelial cells [333].

## 8. Further Direction in the Research on the Role of CC Chemokines in Cancer Processes

The role of individual chemokines in neoplastic processes is very well known. They cause the recruitment of various cells into the cancer niche and cause cancer cell migration and invasion. However, little is known about the interactions between cells in a tumor, particularly the direct or indirect influence of cancer cells on non-cancer cells and the interactions between non-cancer cells. Especially interesting are cancer-cell-induced changes in the expression of chemokines produced by cancer-associated cells, e.g., by TAM [68,69,70,71,257,258], MDSC [67], and CAF [65]. In a tumor, cancer cells are not isolated but interact with cancer-associated cells, and that is why these cells (e.g., TAM, TIL, MDSC, CAF) should be investigated more often to discover new mechanisms in a tumor. Unfortunately, there are few research tools that can be used to show the interactions between cancer-associated cells and the actual cancer cells. The most notable are the co-culture of cancer cells and cancer-associated cells [334] and the use of a conditioned medium from cancer cells to culture non-cancer cells [335,336]. The understanding of those interactions could help foster new therapeutic approaches if the discovered mechanisms are universal in one type of tumor or even common for all neoplastic diseases.

## Figures and Tables

**Figure 1 ijms-21-07619-f001:**
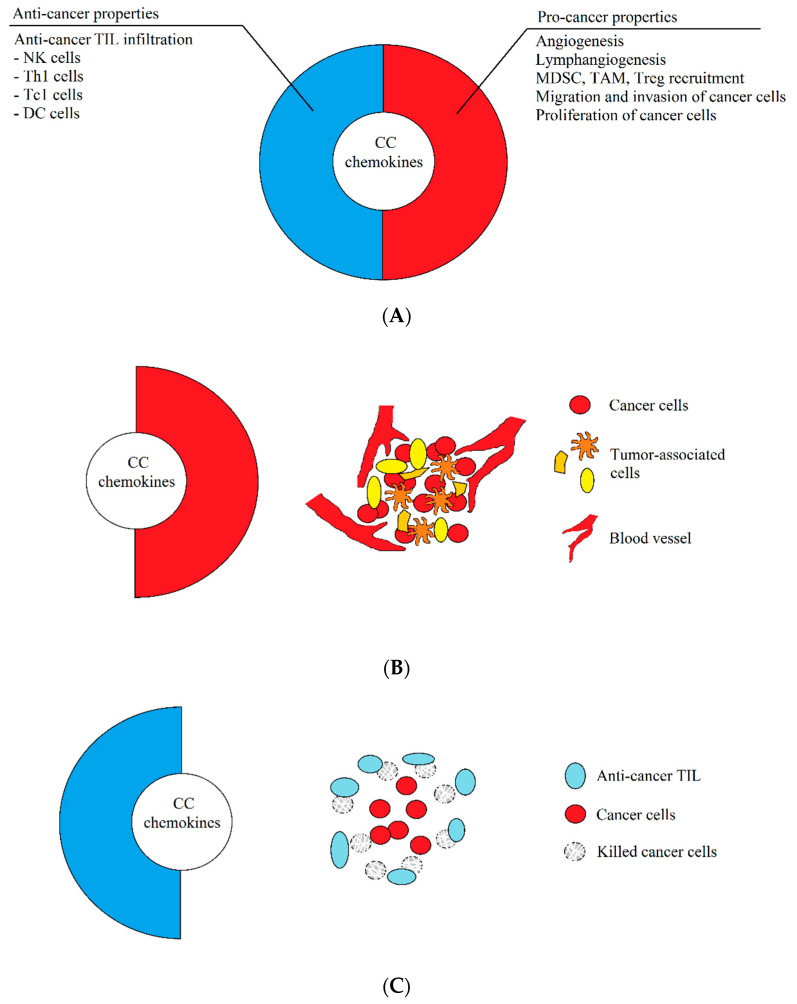
The dual properties of CC chemokines. (**A**) Most, if not all, the chemokines described in this paper have both pro- and anti-cancer properties. The anti-cancer properties consist of the recruitment of anti-cancer tumor-infiltrating lymphocytes (TIL), which infiltrates the tumor and destroys tumor cells. The pro-cancer properties of chemokines, on the other hand, consist in causing angiogenesis and lymphangiogenesis, recruitment of pro-cancer cells supporting the development of the tumor, and the stimulation of proliferation, the induction of migration, and the invasion of cancer cells. (**B**) In a growing tumor, CC chemokines have enhanced pro-cancer properties, while anti-cancer properties are suppressed. As a result, these chemokines participate in the development of a tumor by causing angiogenesis, migration of tumor cells, and recruitment of cells supporting the development of a tumor, which results in the progress of cancer. (**C**) During immunotherapy or an effective anticancer response of the immune system, the same CC chemokines show enhanced anti-cancer properties, which result in the infiltration of a tumor by anti-cancer TIL, which destroy tumor cells. The immune system fights with the tumor, which often leads to recovery.

**Figure 2 ijms-21-07619-f002:**
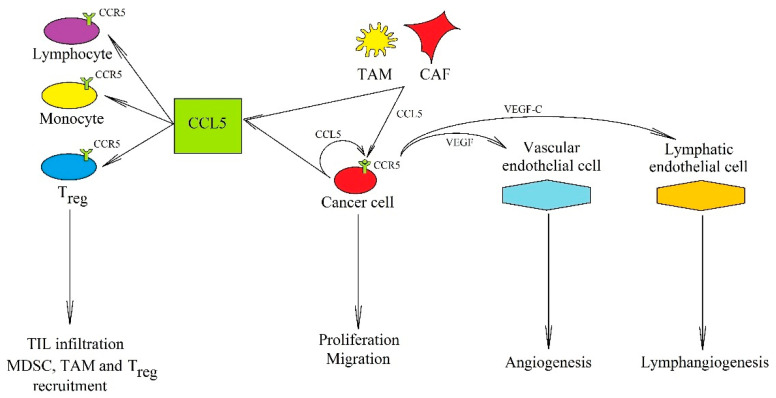
The significance of the CCL5→CCR5 axis in cancer processes. In a tumor, CCL5 is secreted by tumor cells and also by TAM and CAF. Its anti-cancer properties consist of inducing cytotoxic TIL infiltration that increases the anticancer response of the immune system. Its pro-cancer effect is associated with (i) recruitment of cells participating in cancer immune evasion: MDSC, TAM, and T_reg_, (ii) induction of proliferation, migration, and invasion of cancer cells, and (iii) causing an increased production of VEGF, which leads to angiogenesis and lymphangiogenesis.

**Figure 3 ijms-21-07619-f003:**
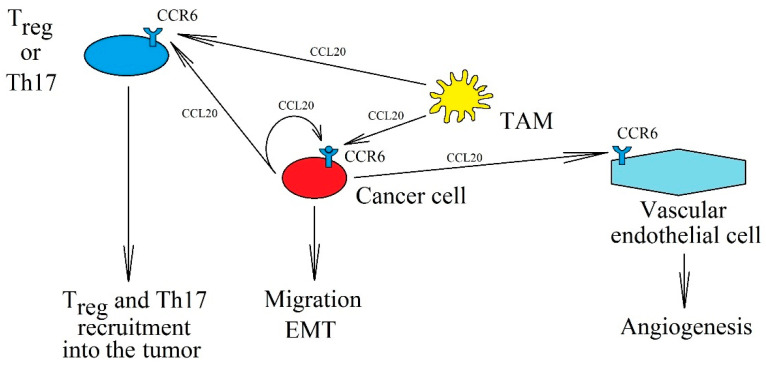
The significance of the CCL20→CCR6 axis in cancer processes. CCL20 is expressed in tumor cells and TAM, which can activate CCR6 on cancer cells in an autocrine manner, causing their migration and EMT. In addition, increased concentration of CCL20 in the cancer microenvironment recruits T_reg_ and Th17 into the tumor niche and causes angiogenesis via CCR6.

**Figure 4 ijms-21-07619-f004:**
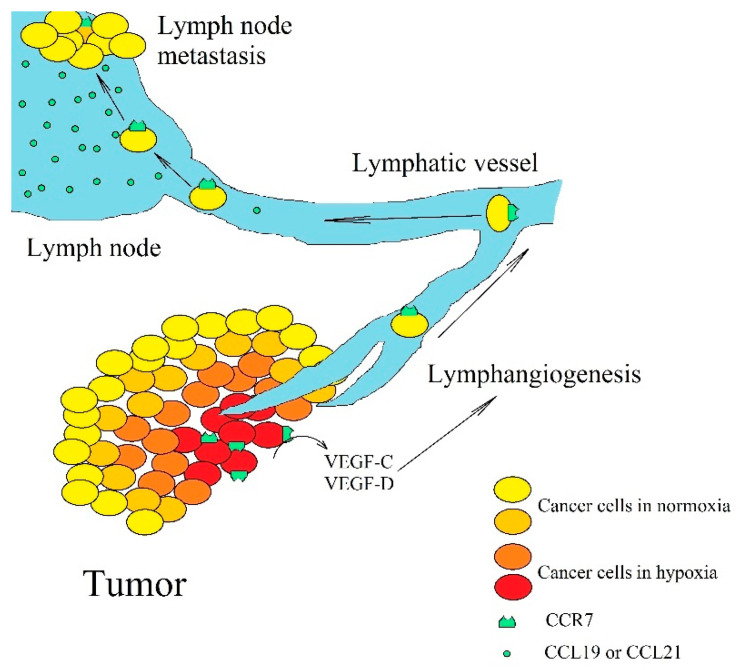
The significance of CCR7 expression in lymph node metastasis. Hypoxia increases CCR7 expression on cancer cells, which leads to an increase in VEGF-C and VEGF-D expression, which in turn, leads to lymphangiogenesis. Then, cancer cells enter lymphatic vessels where they migrate to lymph nodes. As lymph nodes have a high expression of CCL19 and CCL21, cancer cells with CCR7 expression become trapped and form metastasis there.

**Figure 5 ijms-21-07619-f005:**
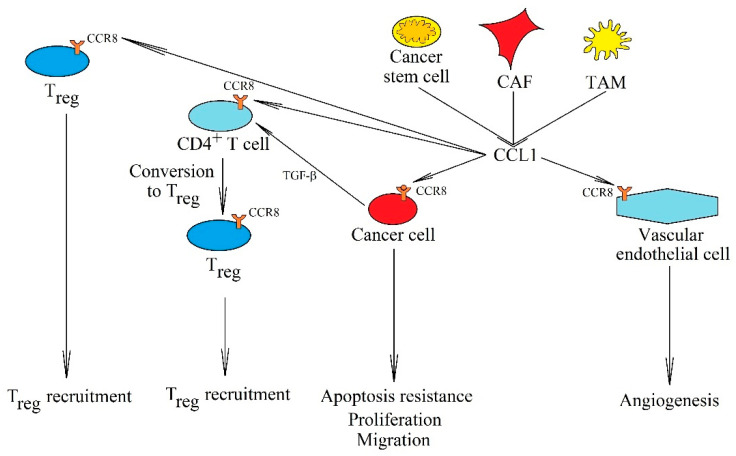
The significance of the CCL1→CCR8 axis in cancer processes. CCL1 is secreted into the cancer microenvironment by cancer stem cells, CAF and TAM. This chemokine has many primary tumor functions. By activating the CCR8 receptor on cancer cells, it causes their proliferation, apoptosis resistance, and migration. It also causes angiogenesis by activating its CCR8 receptor on endothelial cells. Another important function is the recruitment of T_reg_ into the tumor niche and causing the conversion of CD4^+^ T cells into T_reg_.

**Figure 6 ijms-21-07619-f006:**
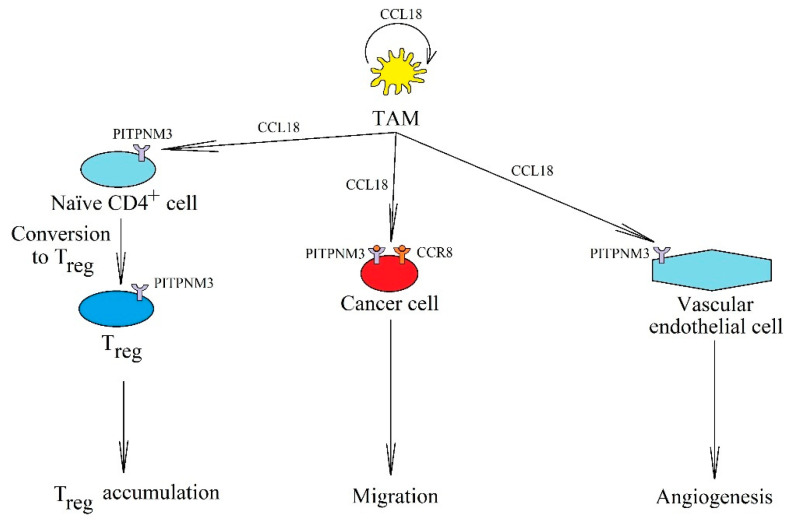
The significance of CCL18 in cancer processes. In a tumor, it is produced and secreted by TAM, where it acts on these cells in an autocrine manner. It also causes cancer cell migration and invasion via receptors PITPNM3 and CCR8. CCL18 is important in angiogenesis by acting on PITPNM3 on endothelial cells. It also causes the recruitment of naïve CD4^+^ T cells into the tumor niche.

**Figure 7 ijms-21-07619-f007:**
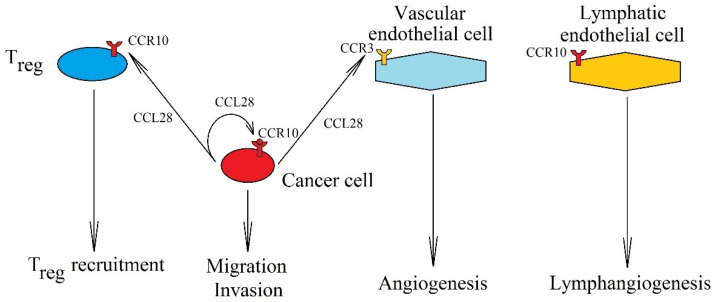
Significance of CCL28 expression increase in cancer processes. Expression of CCL28 in a cancer cell causes the migration and invasion of cancer cells via CCR10. CCL28 is also involved in T_reg_ recruitment into the tumor niche by interacting with receptor CCR10 on these cells. It also causes vascularization of the tumor, causing angiogenesis by activating CCR3 on endothelial cells and lymphangiogenesis by activating CCR10 on lymphatic endothelial cells.

**Table 1 ijms-21-07619-t001:** Influence of increased expression of individual CC chemokines discussed in this review on the prognosis of patients with various cancers according to “The Human Protein Atlas”. (https://www.proteinatlas.org/) [7,8].

Type of Cancer	Receptor										
	CCL1	CCL3	CCL4	CCL5	CCL18	CCL19	CCL20	CCL21	CCL25	CCL27	CCL28
Glioma	N/A	--	--	↓	↓	--	↓*p* = 0.082	N/A	--	N/A	↓
Thyroid cancer	N/A	↑	--	↑	--	--	--	↓	↑*p* = 0.066	N/A	--
Lung cancer	↓*p* = 0.058	↓*p* = 0.089	--	↑	--	--	↓	↓	--	N/A	↓
Colorectal cancer	N/A	--	↑	↑*p* = 0.086	↓*p* = 0.057	↑	--	↓*p* = 0.099	--	N/A	↑
Head and neck cancer	↑	--	↑*p* = 0.070	↑	↑	↑	↓	↑	↑	N/A	↑
Stomach cancer	--	--	--	↑	--	↓*p* = 0.080	--	↓	↓*p* = 0.064	N/A	--
Liver cancer	N/A	--	↑*p* = 0.91	↑*p* = 0.087	--	↑	↓	↑	↑	N/A	↓
Pancreatic cancer	N/A	↑*p* = 0.072	↑*p* = 0.086	↓	↓	↑*p* = 0.083	↓	↑	--	N/A	↓
Renal cancer	N/A	↓	↓	↓	↑	↓	↓	↓	↓	N/A	--
Urothelial cancer	N/A	↑	↑	↑	↓	--	--	↓*p* = 0.089	--	N/A	↑*p* = 0.065
Prostate cancer	N/A	--	--	--	↓	--	--	--	--	N/A	--
Testis cancer	--	↓*p* = 0.093	--	↓*p* = 0.075	--	↓	--	--	--	N/A	--
Breast cancer	N/A	--	↑*p* = 0.060	↑	↑*p* = 0.089	↑	--	↑	↑	N/A	↑
Cervical cancer	↑*p* = 0.065	--	↑	↑	↑	↑	↓	↑	↑	N/A	--
Endometrial cancer	N/A	↑	↑	↑	↑*p* = 0.096	--	↑	↑*p* = 0.055	--	N/A	↓
Ovarian cancer	--	--	--	↑	↑	↑	--	↑	↑	N/A	↓
Melanoma	--	↑	↑	↑	--	↑	--	↓*p* = 0.081	--	N/A	↓

↑ blue background—better prognosis with higher expression of a given chemokine in a tumor; ↓ red background—worse prognosis with higher expression of a given chemokine in a tumor; --—no correlation with higher expression of a given chemokine in a tumor.

**Table 2 ijms-21-07619-t002:** Effects of increased expression of individual CC chemokine receptors discussed in this review on the prognosis of patients with various cancers according to “The Human Protein Atlas” (https://www.proteinatlas.org/) [7,8].

Type of Cancer	Receptor					
	CCR5	CCR6	CCR7	CCR8	CCR9	CCR10
Glioma	↓	↑*p* = 0.076	↓	↓	↑	↓*p* = 0.061
Thyroid cancer	↑	--	--	--	--	↑
Lung cancer	↑	↑	↑	--	↑	--
Colorectal cancer	↑	↑	↑	↑	↑*p* = 0.056	↓*p* = 0.057
Head and neck cancer	↑	↑	↑	↑	↑	↑
Stomach cancer	↑*p* = 0.083	↓	--	↑*p* = 0.080	--	↓
Liver cancer	↑	--	↑	--	--	↓
Pancreatic cancer	--	↑	↑	↑*p* = 0.074	↓*p* = 0.081	↑
Renal cancer	↓	↓	↓	↓	↑	↓
Urothelial cancer	↑	--	↑*p* = 0.079	--	↑	--
Prostate cancer	↑*p* = 0.053	↑	--	--	--	↓
Testis cancer	↓	↓*p* = 0.080	↓	↓	--	--
Breast cancer	↑	↑	↑	--	↑	↑
Cervical cancer	↑	↑	↑	↑	--	↑
Endometrial cancer	↑	--	↑	--	--	↑
Ovarian cancer	↑*p* = 0.062	--	↑	↑	--	--
Melanoma	↑*p* = 0.077	--	↑	↑	--	↓*p* = 0.095

↑ blue background—better prognosis with higher expression of a given chemokine in a tumor; ↓ red background—worse prognosis with higher expression of a given chemokine in a tumor; --—no correlation with higher expression of a given chemokine in a tumor.

**Table 3 ijms-21-07619-t003:** Nomenclature for CC chemokines discussed in this review, including cells recruited into the tumor niche.

Chemokine	Alternative Name of the Chemokine	Receptor	Effect on Recruiting Cells to a Tumor Niche	Effect on Tumor Vascularization	Organ-Specific Metastasis
CCL1	I-309	CCR8	TAM, T_reg_	Angiogenesis	Lymph node
CCL3	MIP-1α	CCR1, CCR5	CAF, MDSC, T_reg_, TIL, Kupffer cells	Angiogenesis	
CCL4	MIP-1β	(CCR1), CCR5	CAF, MDSC, T_reg_, TIL	Angiogenesis, lymphangiogenesis	
CCL5	RANTES	CCR1, CCR3, CCR5	MDSC, TAM, Th17, TIL, T_reg_	Angiogenesis, lymphangiogenesis	
CCL18	PARC, MIP-4	PITPNM3, CCR8	T_reg_	Angiogenesis	
CCL19	ELC	CCR7	TIL, T_reg_	Angiogenesis, lymphangiogenesis	Lymph node
CCL20	LARC	CCR6	T_reg_, Th17	Angiogenesis	Liver
CCL21	SLC	CCR7	TIL, T_reg_	Angiogenesis, lymphangiogenesis	Lymph node
CCL25	TECK	CCR9			Gastrointestinal tract
CCL27	ESkine	CCR10	TIL, Th22	Lymphangiogenesis	Skin
CCL28	MEC	CCR3, CCR10	TIL, T_reg_, cancer-associated stellate cells	Angiogenesis, lymphangiogenesis	

CAF—cancer-associated fibroblasts; MDSC—myeloid-derived suppressor cells; PITPNM3—phosphatidylinositol transfer protein 3; TAM—tumor-associated macrophages; Th17—T helper 17; TIL—anti-cancer tumor-infiltrating lymphocytes; T_reg_—regulatory T cells.

**Table 4 ijms-21-07619-t004:** Receptors for CC chemokines described in this review and their respective ligands and functions in a tumor.

Receptor	Ligands	Effect on Recruiting Cells to a Tumor Niche	Effect on Tumor Vascularization	Organ-Specific Metastasis
CCR5	CCL3, CCL4, CCL5, CCL7, CCL11, CCL14, CCL16	CAF, TIL, MDSC, TAM, T_reg_	Increased VEGF expression, which leads to angiogenesis	
CCR6	CCL20	TAM, Th17, T_reg_	Angiogenesis	Liver
CCR7	CCL19, CCL21	TIL, T_reg_	Increased expression of VEGF-A, VEGF-C, and VEGF-D, which leads to angiogenesis and lymphangiogenesis	Lymph node
CCR8	CCL1, CCL16, CCL18	TAM, T_reg_	Angiogenesis	Lymph node
CCR9	CCL25			Gastrointestinal tract
CCR10	CCL27, CCL28	TIL, T_reg_	Lymphangiogenesis	Skin

CAF—cancer-associated fibroblasts; MDSC—myeloid-derived suppressor cells; TAM—tumor-associated macrophages; Th17—T helper 17; TIL—anti-cancer tumor-infiltrating lymphocytes; T_reg_—regulatory T cells; VEGF—vascular endothelial growth factor.

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
