# Peer review of "CC Chemokines in a Tumor: A Review of Pro-Cancer and Anti-Cancer Properties of Receptors CCR5, CCR6, CCR7, CCR8, CCR9, and CCR10 Ligands"

_ijms, 2020, doi:10.3390/ijms21207619_

Round 1

Reviewer 1 Report

The review is a catalogue of data regarding CC chemokines and cancer without any perspective, or adoption of a concept that progresses understanting on the topic. This is reflected in the "empty'' summary (it simply reports what it will be discussed) and the almost "empty" last section 8: understanding wanted.

Author Response

Rev.1

The review is a catalogue of data regarding CC chemokines and cancer without any perspective, or adoption of a concept that progresses understanding on the topic. This is reflected in the "empty'' summary (it simply reports what it will be discussed) and the almost "empty" last section 8: understanding wanted.

Introduction and the final section of the artice have been extended.

In the introduction we have added a section to make sure that the readers pay special attention to the dual properties of each chemokine, i.e. simultaneous pro- and anti-cancer properties. Understanding this nature of chemokines will allow us to understand the functioning of cancer tumors and the effects of many therapies. Experimental studies often report conflicting results, indicating therapies which should increase or decreasing the expression a given chemokine. Our idea is different: an increase in the expression of a given chemokine should be followed by enhancing its anti-cancer properties. However, to do this we need to know the anti-cancer and pro-cancer properties of chemokines, as we have included in our review.

Reviewer 2 Report

The authors present a detailed manuscript, discussing the role in cancer of several CC chemokine receptors and their associated ligands.  The work is exhaustive, discussing multiple cancer types and the role of chemokines/receptors in each type, with often different characteristics:  for any given chemokine/receptor pair there may be cancer types where the pair appears to have anti-cancer properties, and cancer types where the same chemokine/receptor pair appear to have pro-cancer properties. In some cases, the chemokines can recruit cells that help control the tumor (such as CCL20/CCR6), in other cases Treg cells are recruited, which assists in tumor immune evasion (also CCL20/CCR6).

            The manuscript is extensively referenced, so the reader can find the original article about the particular chemokine or receptor of interest.  It is also well-written, with essentially no grammatical errors or even stilted sentences.

            This manuscript should be quite useful to someone who wants the details of the role of a particular CC chemokine in a particular cancer or in a wide array of cancers.  The Tables nicely lay out each chemokine and (separately) each receptor, and its general role in a variety of cancers (i.e. whether its presence leads to a better or worse prognosis when upregulated).  Further, there are several useful figures, individually detailing the role of a single chemokine/receptor pair on their variety of targets related to cancer and cancer immunology.

            My only complaint is that it might be nice if the authors explain the implications sometimes of the huge number of facts that they are presenting.  For example, on line 243, the say that under certain conditions CCL19 and CCL21 may participate in Treg recruitment, but that is the end of the section.  So therefore, what?  I assume they are saying that this means that under those conditions, CCL19 and CC21 may participate in immune suppression of the tumor. 

I have a very few minor suggestions:

  1. The first sentence in the abstract could be made more precise.  It says, “CC chemokines (or β-chemokines) are 28 chemotactic cytokines… that play an important role in immune system cells …”   I think it is not quite correct to say that CC chemokines play a role in the cells themselves.  The chemokines play a role in activation, chemotaxis, and homing of immune system cells. 
  2. Tables 1 and 2: Why does a *down* arrow indicate *better* prognosis with upregulation of that chemokine/receptor, while an *up* arrow indicates *worse* prognosis?  Shouldn’t an up arrow denote better prognosis?  Also, the authors should define what is orange and what is blue.
  3. Tables 3 and 4 have alignment issues, where column 1 doesn’t align with column 2.
  4. Line 89 Figure legend for Figure 1: “…properties consist in inducing…” should probably be “…properties consist OF inducing…”
  5. Line 123: Typo. Combadiere et al. show THAT CCL3 and CCL4…
  6. Line 186 and 272: Need more space for arrow between chemokine and receptor.
  7. Line 329: The use of “however”.  This is probably a generational divide, but I would say that this is not the proper use of “however” in the middle of a sentence.  The sentence should end, then a new sentence should start with “however”:  ….its participation in angiogenesis.  However, this chemokine may…
  8. Lines 344 and 374 nearly duplicate each other.  They are not quite saying the same thing, so perhaps the authors do intend to keep both sentences.
  9. Line 405: Need EMT in list of Abbreviations.  Possibly could use TADC in list of abbreviations.

Author Response

Rev.2

The authors present a detailed manuscript, discussing the role in cancer of several CC chemokine receptors and their associated ligands.  The work is exhaustive, discussing multiple cancer types and the role of chemokines/receptors in each type, with often different characteristics:  for any given chemokine/receptor pair there may be cancer types where the pair appears to have anti-cancer properties, and cancer types where the same chemokine/receptor pair appear to have pro-cancer properties. In some cases, the chemokines can recruit cells that help control the tumor (such as CCL20/CCR6), in other cases Treg cells are recruited, which assists in tumor immune evasion (also CCL20/CCR6).

            The manuscript is extensively referenced, so the reader can find the original article about the particular chemokine or receptor of interest.  It is also well-written, with essentially no grammatical errors or even stilted sentences.

            This manuscript should be quite useful to someone who wants the details of the role of a particular CC chemokine in a particular cancer or in a wide array of cancers.  The Tables nicely lay out each chemokine and (separately) each receptor, and its general role in a variety of cancers (i.e. whether its presence leads to a better or worse prognosis when upregulated).  Further, there are several useful figures, individually detailing the role of a single chemokine/receptor pair on their variety of targets related to cancer and cancer immunology.

            My only complaint is that it might be nice if the authors explain the implications sometimes of the huge number of facts that they are presenting.  For example, on line 243, the say that under certain conditions CCL19 and CCL21 may participate in Treg recruitment, but that is the end of the section.  So therefore, what?  I assume they are saying that this means that under those conditions, CCL19 and CC21 may participate in immune suppression of the tumor. 

 A sentence explaining the importance of recruiting Treg has been added

I have a very few minor suggestions:

  1. The first sentence in the abstract could be made more precise.  It says, “CC chemokines (or β-chemokines) are 28 chemotactic cytokines… that play an important role in immune system cells …”   I think it is not quite correct to say that CC chemokines play a role in the cells themselves.  The chemokines play a role in activation, chemotaxis, and homing of immune system cells. 

The sentence has been corrected. We have also extended the whole introduction and presented the purpose of writing this review.

  1. Tables 1 and 2: Why does a *down* arrow indicate *better* prognosis with upregulation of that chemokine/receptor, while an *up* arrow indicates *worse* prognosis?  Shouldn’t an up arrow denote better prognosis?  Also, the authors should define what is orange and what is blue.

This is an error. Up arrow denotes better prognosis. The colors are explained now.

  1. Tables 3 and 4 have alignment issues, where column 1 doesn’t align with column 2.

There is a different number of verses in each cell which makes the tables unreadable. A line has been added between the verses.

  1. Line 89 Figure legend for Figure 1: “…properties consist in inducing…” should probably be “…properties consist OF inducing…”

Corrected.

  1. Line 123: Typo. Combadiere et al. show THAT CCL3 and CCL4…

Corrected.

  1. Line 186 and 272: Need more space for arrow between chemokine and receptor.

This is an error that occurred after the text was converted into PDF format by the program on the editorial site. In the DOC version, the arrows look correct. We have changed the arrows to those that have converted well. But I cannot guarantee that they will convert well. If not, I will put a space there.

  • Line 329: The use of “however”.  This is probably a generational divide, but I would say that this is not the proper use of “however” in the middle of a sentence.  The sentence should end, then a new sentence should start with “however”:  ….its participation in angiogenesis.  However, this chemokine may…

Corrected

  1. Lines 344 and 374 nearly duplicate each other.  They are not quite saying the same thing, so perhaps the authors do intend to keep both sentences.

The first sentence was deleted because it was in a paragraph about the physiological role of CCR10 .

  1. Line 405: Need EMT in list of Abbreviations.  Possibly could use TADC in list of abbreviations.

EMT is now in abbreviations

TADC appears in the review only once. In order to limit the number of abbreviations, this section does not contain abbreviations appearing only once or in one paragraph only.

Reviewer 3 Report

This manuscript, written by Dr. Jan Korbecki et al., with the title of “CC chemokines in a tumor. A review of pro-cancer 2 and anti-cancer properties of receptors CCR5, CCR6, 3 CCR7, CCR8, CCR9, and CCR10 ligands” is a review that focuses on the role of CC chemokines and their receptors in the tumor biology. The role of chemokines in cancer is know but it is a very complex situation as there are many factors, and each chemokine/receptor can have several roles depending on the type of cancer and location. This manuscript aims to summary the most relevant information.

After a quick introduction the authors show the clinicopathological correlations of these chemokines and receptors in several types of cancer, based on the information present in the protein atlas webpage. Then, the authors make a very thorough descriptions of chemokines and receptors in several sections including CCR5 ligands, CCR6/CCL20, CCR7/CCL19/CCL21, CCR8, CCR9/CCL15, and CCR10/CCL28, CCL27. Finally, the authors make a conclusion.

This review manuscript is well written, it is easy to read, there are enough figures and tables, and contain enough references (n=269). The figures are genuine and very informative. Before publication, the authors could address some of the minor comments.

Minor comments:

1) Please be careful with the Tables 1-4. In the current manuscript the tables are cut in several pages so one cannot read them properly (this may be changed by the editorial office in the final version of the manuscript). In the Table 3 and 4 there a few horizontal lines, and I had difficulty in knowing which part of the information correspond to each chemokine.

2) Page 4, line 78. “The CCL5->CCR5 axis also induces chemotaxis and increases the cytotoxic properties of tumor-infiltrating lymphocytes (TIL), such as CD4+ lymphocytes, CD8+ 78 lymphocytes, dendritic cells, and natural killers (NK) cells [17-23].” The authors could specify which specific subtypes of lymphocytes are involved. For example, for CD4+ lymphocytes could be Th, TFH and Tregs; for CD8+ could be Tc. All have different functions in the tumor biology. If the CCL5-CCR5 is due to be targeted by drugs, one should be careful of the effect on the cells. The authors may expand this section with more information, if it is available in the literature (of note, part of this information is present in Figure 1 but not in the text).

3) This review contains a lot of information. The authors could read it carefully again and try to simplify a little and remove redundant information.

4) The authors do not mention neoplasia of the hematopoietic and lymphoid tissues. For example, in one of the most frequent non-Hodgkin lymphomas, the diffuse large b-cell lymphoma (DLBCL), the chemokines have significant role in the prognosis of the patients.

5) The authors could also add more information regarding HIV and/or other immune deficiencies in the context of cancer.

6) If the authors have access to histological pictures, the authors could add some examples of the protein expression of some of the chemokines/receptors.

7) At the conclusion section, the authors could highlight the most relevant points in a few sentences.

Author Response

Rev.3

This manuscript, written by Dr. Jan Korbecki et al., with the title of “CC chemokines in a tumor. A review of pro-cancer 2 and anti-cancer properties of receptors CCR5, CCR6, 3 CCR7, CCR8, CCR9, and CCR10 ligands” is a review that focuses on the role of CC chemokines and their receptors in the tumor biology. The role of chemokines in cancer is know but it is a very complex situation as there are many factors, and each chemokine/receptor can have several roles depending on the type of cancer and location. This manuscript aims to summary the most relevant information.

After a quick introduction the authors show the clinicopathological correlations of these chemokines and receptors in several types of cancer, based on the information present in the protein atlas webpage. Then, the authors make a very thorough descriptions of chemokines and receptors in several sections including CCR5 ligands, CCR6/CCL20, CCR7/CCL19/CCL21, CCR8, CCR9/CCL15, and CCR10/CCL28, CCL27. Finally, the authors make a conclusion.

This review manuscript is well written, it is easy to read, there are enough figures and tables, and contain enough references (n=269). The figures are genuine and very informative. Before publication, the authors could address some of the minor comments.

Minor comments:

1) Please be careful with the Tables 1-4. In the current manuscript the tables are cut in several pages so one cannot read them properly (this may be changed by the editorial office in the final version of the manuscript). In the Table 3 and 4 there a few horizontal lines, and I had difficulty in knowing which part of the information correspond to each chemokine.

The tables have been modified.

2) Page 4, line 78. “The CCL5->CCR5 axis also induces chemotaxis and increases the cytotoxic properties of tumor-infiltrating lymphocytes (TIL), such as CD4+ lymphocytes, CD8+ 78 lymphocytes, dendritic cells, and natural killers (NK) cells [17-23].” The authors could specify which specific subtypes of lymphocytes are involved. For example, for CD4+ lymphocytes could be Th, TFH and Tregs; for CD8+ could be Tc. All have different functions in the tumor biology. If the CCL5-CCR5 is due to be targeted by drugs, one should be careful of the effect on the cells. The authors may expand this section with more information, if it is available in the literature (of note, part of this information is present in Figure 1 but not in the text).

The section has been changed according to the Reviewer's comments.

3) This review contains a lot of information. The authors could read it carefully again and try to simplify a little and remove redundant information.

The aim of our review is to create a compendium of knowledge on the importance of individual chemokines in cancer processes. After reading our work, the scientist should not look for experimental papers published before our review. That is the reason why we tried to include all significant data. We have alread tried to limit the length of the paper, and so we focused on intercellular signaling and not on intracellular signaling in a cancer cell.

4) The authors do not mention neoplasia of the hematopoietic and lymphoid tissues. For example, in one of the most frequent non-Hodgkin lymphomas, the diffuse large b-cell lymphoma (DLBCL), the chemokines have significant role in the prognosis of the patients.

We have added fragments on non-Hodgkin lymphomas and DLBCL

5) The authors could also add more information regarding HIV and/or other immune deficiencies in the context of cancer.

We have added information on the oncogenic viruses (HHV-8, HPV and EBV) on the discussed chemokines and their receptors. We have not found anything on HIV. HTLV-1 is already discussed in the initial version of the manuscript.

6) If the authors have access to histological pictures, the authors could add some examples of the protein expression of some of the chemokines/receptors.

We have a plan to research two chemokine axis in cancers, based on the samples we have already collected. The experimental stage of the research is planned to start in a couple of months. Histological pictures will included in our experimental works.

7) At the conclusion section, the authors could highlight the most relevant points in a few sentences.

Conclusion has been extended.